# Towards An Objective Evaluation of the Trustworthiness of Classifiers

## Abstract

With the widespread deployment of AI models in applications that impact human lives, research on model trustworthiness has become increasingly important, as a result of which model effectiveness alone (measured, e.g., with accuracy, F1, etc.) should not be the only criteria to evaluate predictive models; additionally the trustworthiness of these models should also be factored in. It has been argued that the features deemed important by a black-box model should be aligned with the human perception of the data, which in turn, should contribute to increasing the trustworthiness of a model. Existing research in XAI evaluates such alignments with user studies - the limitations being that these studies are subjective, difficult to reproduce, and consumes a large amount of time to conduct. We propose an evaluation framework, which provides a quantitative measure for trustworthiness of a black-box model, and hence, we are able to provide a fair comparison between a number of different black-box models. Our framework is applicable to both text and images, and our experiment results show that a model with a higher accuracy does not necessarily exhibit better trustworthiness.

## 1 Introduction

Owing to the success and promising results achieved for data-driven (deep) approaches for supervised learning, there has been a growing interest in the AI community to apply such models in domains such as healthcare (Asgarian et al., 2018; Spann et al., 2020; Yasodhara et al., 2020), criminal justice (Rudin, 2019) and finance (Dixon et al., 2020). As ML models become embedded into critical aspects of decision making, their successful adoption depends heavily on how well different stakeholders (e.g. user or developer of ML models) can understand and trust their predictions. As a result, there has been a recent surge in making ML models worthy of human trust (Wiens et al., 2019), and researchers have proposed a variety of methods to explain ML models to stakeholders (Bhatt et al., 2020), with examples such as DARPA's Explainable AI (XAI) initiative (Gunning et al., 2019) and the 'human-interpretable machine learning' community (Abdul et al., 2018).

Although standard evaluation metrics exist to evaluate the performance of a predictive model, there is no consistent evaluation strategy for XAI. Consequently, a common evaluation strategy is to show individual, potentially cherry-picked, examples that look reasonable (Murdoch et al., 2019) and pass the first test of having 'face-validity' (Doshi-Velez & Kim, 2018). Moreover, evaluating the ability of an explanation to convince a human is different from evaluating its correctness, e.g., while Petsiuk et al. (2018) believe that keeping humans out of the loop for evaluation makes it more fair and true to the classifier's own view on the problem rather than representing a human's view, Gilpin et al. (2018) explain that a non-intuitive explanation could indicate either an error in the reasoning of the predictive model, or an error in the explanation producing method.

Visual inspection on the plausibility of explanations, such as anecdotal evidence, cannot make the distinction as to whether a non-intuitive explanation is the outcome of an error in the reasoning of the predictive model, or that it is an error that could be attributed to an explanation generating model itself. Zhang et al. (2019) identify such visual inspections as one of the main shortcomings when evaluating XAI and state that checking whether an explanation "looks reasonable" only evaluates the accuracy of the black box model and is not evaluating the faithfulness of the explanation. These commentaries relate to the inherent coupling of evaluating the black box model's predictive accuracy

with explanation quality. As pointed out by Robnik-Sikonja & Bohanec (2018), the correctness of an explanation and the accuracy of the predictive model may be orthogonal.

Although the correctness of the explanation is independent of the correctness of the prediction, visual inspection cannot distinguish between the two. Validating explanations with users can unintentionally combine the evaluation of explanation correctness with evaluating the correctness of the predictive model. Synthetic datasets are useful for evaluating explanations for black box models Oramas et al. (2019). By designing a dataset in a controlled manner, it should be possible to argue, with a relatively high confidence, that a predictive model should reason in a particular way; a set of 'gold' explanations can thus be created in a controlled manner using a data generation process. Subsequently, the agreement of the generated explanations with these true explanations can be measured. For example, Oramas et al. (2019) generate an artificial image dataset of flowers, where the color is the discriminative feature between classes.

Our work **compares multiple underlying predictive models in terms of trustworthiness, rather than the XAI methods themselves.** Evaluating whether the features deemed to be important by a predictive model conform with those by a human is an intrinsically human-centric task that ideally requires human studies. However, performing such studies multiple times during the model development phase is not feasible. To this end, the major contributions of our work are as follows.

**Our Contributions.** First, we generate a synthetic dataset and its associated ground-truth explanations for a multi-objective image classification task. We also manually create the ground-truth explanations for two image classification datasets, namely the MNIST '3 vs. 8' classification and the Plant-Village (Mohanty et al., 2016) disease classification tasks, and for our text experiments we make use of a dataset of legal documents with existing ground-truth explanation units (Malik et al., 2021) (dataset and code will be released).

Second, we propose a general framework to quantify the trustworthiness of a black-box model. Our approach is agnostic to both the explanation methodology and the data modality. In our experiments, we compare the performance of predictive models both in terms of **effectiveness and trustworthiness** on synthetic and real-world datasets using data from two different modalities - image and text.

## 2 RELATED WORK

Several works have used synthetic datasets for evaluating XAI algorithms. Liu et al. (2021) released the XAI-BENCH - a suite of synthetic datasets along with a library for benchmarking feature attribution algorithms. The authors argue that their synthetic datasets offer a wide variety of parameters which can be configured to simulate real-world data and have the potential to identify subtle failures, such as the deterioration of performance on datasets with high feature correlation. They give examples of how real datasets can be converted to similar synthetic datasets, thereby allowing XAI methods to be benchmarked on realistic synthetic datasets.

Oramas et al. (2019) introduce an8Flower, a dataset specifically designed for objective quantitative evaluation of methods for visual explanation. They generate two synthetic datasets, 'an8Flower-single-6c' and 'an8Flower-double-12c', with 6 and 12 classes respectively. In the former, a fixed single part of the object is allowed to change color. This color defines the classes of interest. In the latter, a combination of color and the part on which it is located defines the discriminative feature. After defining these features, they generate masks that overlap with the discriminative regions. Then, they threshold the heatmaps at given values and measure the pixel-level intersection over union (IoU) of a model explanation (produced by the method to be evaluated) with respect to these masks. We argue that the importance of each pixel as outputted by the XAI model is different, and that this information is not captured by a simple technique, such as the pixel-level IoU of a model's explanations relative to the ground-truth explanation masks. In our work, we propose two new metrics for **evaluating predictive models** in terms of their trustworthiness.

Some argue in favor of automated metrics where no user involvement is needed, e.g., in the context of usability evaluation in the Human Computer Interaction (HCI) community, Greenberg & Buxton (2008) argue that there is a risk of executing user studies in an early design phase, since this can quash creative ideas or promote poor ideas. Miller et al. (2017) therefore argues that proxy studies are especially valid in early development. Qi et al. (2021) indicate that "evaluating explanations

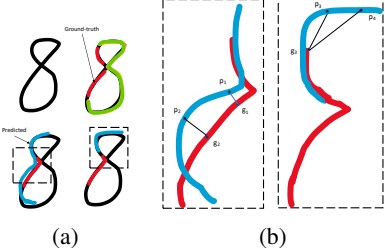

(a)              (b)

Figure 1: A schematic description of the idea leveraged by the proposed trustworthiness metric on a sample image of a '3 vs. 8' classification task. **a**): Overlayed ground-truth (red) and the predicted feature weights (blue) of explanations obtained for two different black-box models. **b**): A zoomed-in view of selected regions from these two explanations showing the closest ground-truth from a pair of predicted points $p_1$ and $p_2$ (left), and $p_3$ and $p_4$ on the right. The trustworthiness of the model on the left is higher than that of the right, because the explanations for the former closely matches the human perceived discriminating feature (shown in red). It can be seen that the sum of the distances between the two line segments for the figure on the right is higher than that of the left. This results in a higher quantitative value of the metric for the image on the left.

objectively without a human study is also important because simple parameter variations can easily generate thousands of different explanations, vastly outpacing the speed of human studies".

User studies are not only relatively hard to run, they may be of limited value (Poursabzi-Sangdeh et al., 2021) and may suffer from confirmation bias (Wang et al., 2019). Rosenfeld (2021) present four objective XAI measures to quantify XAI effectiveness either on their own or in conjunction with user studies: '**D**', the performance difference between the agent's model and the performance of the logic presented as an explanation, '**R**', the number of rules in the agent's explanation, '**F**', the number of features used to construct the explanation and '**S**', the stability of the agent's explanation.

Amiri et al. (2020) proposed a methodology to create a synthetic dataset representing ground-truth explanations (GTE), and used it for evaluating LIME. Our work is different in the sense that we focus on evaluating predictive models rather than the XAI method. We, therefore, define GTE in terms of human perception of important features, e.g. the patches on the leaves of plants that identify a particular disease. Yang & Kim (2019) release a carefully crafted semi-natural image dataset with ground-truth for evaluating interpretability of methods. They also propose three complementary metrics to evaluate interpretability - a) Model Contrast Score (MCS), which measures the performance of interpretability methods across models, b) Input Dependence Rate (IDR), which accounts for when an interpretability method should "react" to two different inputs, and c) Input Independence Rate (IIR), which is concerned with when an interpretability method should not "react" to two different inputs. Our work, however, does not focus on evaluating XAI methods in terms of identifying false positives (features that are incorrectly attributed as important) or false negatives. We rather focus on **evaluating predictive models with respect to their trustworthiness, keeping the XAI method constant**. Yang & Kim (2019) train a model in a controlled manner and establish what regions of the input the model *is* and *isn't* attending to. The XAI methods are then evaluated based on whether they're able to capture such important and non-important features in their explanations of the model. In our work, however, we define GTE in terms of human perception of important features, and then compare different predictive models in terms of the extent to which they agree to this ground-truth for making their predictions.

## 3   TRUSTWORTHINESS EVALUATION OF MODELS

**Constructing an objective ground-truth of trustworthiness.** Existing studies have mainly used the trustworthiness measure to compute the fitness of an XAI model itself, e.g., Chen et al. (2018) compare the stability of feature weight distributions across samples of data instances computed by different explanation methodologies, and report that L2X leads to the most stable results. The novelty of our work lies in objectively evaluating the trustworthiness of black-box models. To this end, our evaluation framework first requires a manually created ground-truth defined for a small number of data instances, e.g., for images, this ground-truth is in the form of regions (constituted of

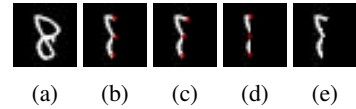

(a)    (b)    (c)    (d)    (e)

Figure 2: An illustration of how we obtain the ground-truth explanations from the manual assessments provided by three different annotators for the 3-vs-8 classification task. Each annotator required to select 3 control points (shown as red dots), based on which a region was selected as the ground-truth. The final ground-truth annotation is an average image of the three individual ones. The red dots are magnified in the figure so that they can be seen conveniently. The images, in order from left to right, correspond to - a) Original Image, b) Annotator 1, c) Annotator 2, d) Annotator 3, e) Ground-truth.

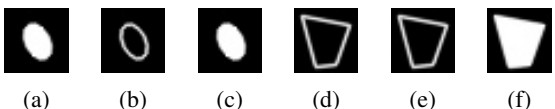

(a)      (b)      (c)      (d)      (e)      (f)

Figure 3: Synthetic data and corresponding ground-truth explanations for a multi-objective image classification task. **Task 1**: Shape classification and **Task 2**: To classify if filled or not. The images, in order from left to right, correspond to - a) Generated image where Shape='Ellipse' and Filled='1', b) Ground-truth for the shape recognition task, c) Ground-truth for recognizing whether filled, d) Generated image where Shape='Quad' and Filled='0', e) Ground-truth for the shape recognition task, and f) Ground-truth for recognizing whether filled.

pixels) within an image indicative of discriminative features and for texts, it is in the form of a set of sentences.

**A Distance Weighted Overlap Measure.** We present our approach by referring to data units as pixels being associated with a 2D coordinate. However, the same principle also applies for text where each explanation unit (a word or a sentence etc.) is specified by a single dimension (namely its offset or index in the text). Let $\theta : X \mapsto \mathbb{Z}_k$ be a data-driven black-box model that classifies an input $\mathbf{x} \in \mathbb{R}^{h \times w}$ (or simply $\in \mathbb{R}^w$ for text) to one of the $k$ classes, where $h$ and $w$ are the 2D coordinates of an input feature (for an image, and $w$ is an integer offset of a word or a sentence in text). Let us denote the objective trustworthiness ground-truth of an input $\mathbf{x}$ as $\tau(\mathbf{x})$ (this ground-truth is either manually annotated or synthetically generated, as described later in Section 4).

The ground-truth $\tau(\mathbf{x})$ of an input $\mathbf{x}$ is a binary matrix of the same dimensions as that of $\mathbf{x}$, or in other words, $\tau(\mathbf{x}) \in \{0, 1\}^{h \times w}$. Let the local feature weight predictions outputted by some XAI method be denoted by $\phi_{\mathbf{x},\theta}$. For the particular case of image classification, the weight distribution corresponds to a pixel-level importance and hence is of the same dimension as that of $\mathbf{x}$, i.e., $\phi_{\mathbf{x},\theta} \in \mathbb{R}^{h \times w}$. Similar to Ribeiro et al. (2016), we select a subset of the most important features from the predicted feature weight distribution matrix. A convenient way of doing this is to select the top-$(100 \times k)$ percentile weights, where $k \in [0, 1]$ is a parameter (we denote this set with the simple notation $\phi_k$). The set $\phi_k$ thus constitutes tuples of the form $(h, w, c)$, where $c \in [0, 1]$ is the importance of the feature at index $(h, w)$.

The problem with a simple overlap (e.g., Jaccard-based) metric (Oramas et al., 2019) is that it cannot take into account the relative distances between the predicted and the ground-truth features, and as a result, may end up penalizing a predicted feature within a near vicinity of a ground-truth one. Figure 1 illustrates the idea, where we revisit the case-study of a '3 vs. 8' classifier; the human perceived discriminating feature between a '3' and an '8' is the left part of the looped curve of an '8', the absence of which transforms it to a '3'. Figure 1b shows that the model on the left results in a more trustworthy prediction, and this is captured by the sum of the distances of a predicted (to be an important feature) point from its nearest ground-truth point.

**Precision Measure.** The first metric that we propose is precision-based, where we measure the fraction of features that are deemed important both by the predictive model and by humans. More concretely, we start by dividing $\phi_k$ and $\tau(\mathbf{x})$ into windows of shape $m \times m$, to get $\phi_k^m$ and $\tau(\mathbf{x})^m$ respectively. For each predicted window, we find its closest window in the ground-truth set, and then add up the similarities (inverse distances). We also make use of the feature importance values to compute a weighted average to take into account the confidence of the predictions of feature

**Algorithm 1:** Algorithm to compute trustworthiness metrics - TWP and TWR.

**Input:** Ground-truth explanations for an evaluation set $X_e$, i.e., $\{\tau(\mathbf{x}), \mathbf{x} \in X_e\}$:
**Input:** A black-box model $\theta$, and a local explanation methodology $\phi$, e.g., LIME or SHAP etc.
**Input:** $\forall(z, i, j)\ s_{i,j}^z$: The importance of pixel $(i, j)$ of image $z \in X_e$, obtained by an explanation method.
**Input:** $k \in [0, 1]$: The top-$(k \times 100)$ percentile of the feature weights to be used to define the predicted set of important features.
**Input:** $m$: size of window for trustworthiness metrics computation
**Output:** TWP, TWR
**begin**
    $\text{TWP}_{all} \leftarrow 0;\ \text{TWR}_{all} \leftarrow 0.$
    **foreach** $\mathbf{x}$ *in* $X_e$ **do**
        $\phi_{\mathbf{x},\theta} \leftarrow$ The local explanation for the black-box model $\theta$ on $\mathbf{x}$.
        Sort each element $(h, w, c) \in \phi_{\mathbf{x},\theta}$ by descending values of $c$, and retain the top-$(k \times 100)$ percentile; call this $\phi_k$.
        $\text{TWP} \leftarrow 0;\ \text{TWR} \leftarrow 0.$
        $Z_p \leftarrow 0;\ Z_r \leftarrow 0$ // Normalization factors for TWP and TWR
        $\phi_k^m \leftarrow$ Set of all $m \times m$ windows in $\phi_k$
        $\tau(\mathbf{x})^m \leftarrow$ Set of all $m \times m$ windows in $\tau(\mathbf{x})$
        **foreach** $p$ in $\phi_k^m$ **do**
            **if** $\exists(h, w, c) : (h, w, c) \in p \wedge c \neq 0$ **then**
                $g \leftarrow$ Closest window in $\tau(\mathbf{x})^m$ to the window $p$
                $c \leftarrow \frac{\sum_{(h,w,c) \in p} c}{|p|}$
                $\text{TWP} \leftarrow \text{TWP} + c \cdot e^{-d(p,g)}$
                $Z_p \leftarrow Z_p + c$

        **foreach** $g$ in $\tau(\mathbf{x})^m$ **do**
            **if** $\exists(h', w') : (h', w') \in g \wedge \tau(\mathbf{x})_{h',w'} \neq 0$ **then**
                $p \leftarrow$ Closest window in $\phi_k^m$ to the window $g$
                $c' \leftarrow \frac{\sum_{(h,w,c) \in p} c}{|p|}$
                $\text{TWR} \leftarrow \text{TWR} + c' \cdot e^{-d(g,p)}$
                $Z_r \leftarrow Z_r + c'$

        $\text{TWP} \leftarrow \text{TWP}/Z_p;\ \text{TWR} \leftarrow \text{TWR}/Z_r$
        $\text{TWP}_{all} \leftarrow \text{TWP}_{all} + \text{TWP}$
        $\text{TWR}_{all} \leftarrow \text{TWR}_{all} + \text{TWR}$
    **return** $\text{TWP}_{all}/|X_e|, \text{TWR}_{all}/|X_e|$. // Average over the evaluation set

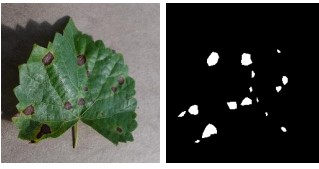

Figure 4: Sample annotated ground-truth explanation from plant disease classification task.

importance. Formally speaking,

$$\text{TWP}_m = \frac{\sum\limits_{p \in \phi_k^m} c \cdot e^{-d(p,g)}}{\sum\limits_{p \in \phi_k^m} c}, \text{ where } g \text{ is the closest window in } \tau(\mathbf{x})^m \text{ to the window } p, \quad (1)$$

where $d((p, g)$ is the average distance between the set of points in $p$ and those in $g$ and $c$ is the average feature importance value of window $p$. We call this metric TWP (abbv. for **TrustWorthiness Precision**) because of its similarity with the set-based definition of the precision measure. The denominator denotes the set of most important (predicted) features, whereas the numerator denotes a weighted correctness measure (similar to the true-positives).

**Recall Measure.** The second metric that we propose measures the fraction of human perceived important features that are deemed important by the predictive model under consideraion. For measuring soft recall, we start from each window in the ground-truth and find its dual (the closest window in the predicted set). The dual has the importance score $c'$. We then aggregate the similarities

| Dataset | XAI Method | FF | | | | | | CONV | | | | | |
|---|---|---|---|---|---|---|---|---|---|---|---|---|---|
| | | Fidelity | F1 | TWP$_1$ | TWR$_1$ | TWP$_4$ | TWR$_4$ | Fidelity | F1 | TWP$_1$ | TWR$_1$ | TWP$_4$ | TWR$_4$ |
| Syn (shape) | SHAP | 0.97 | 0.9867 | **_0.7654_** | **_0.7685_** | **_0.1303_** | **_0.1370_** | 0.36 | 0.9933 | 0.6749 | 0.7586 | 0.1242 | 0.1342 |
| | LIME | 0.66 | | **0.7672** | **0.7603** | **0.1303** | **0.1222** | 0.67 | | 0.7442 | 0.7297 | 0.1276 | 0.1167 |
| Syn (filled) | SHAP | 0.51 | 1.0000 | **_0.9418_** | **_0.8017_** | **_0.1297_** | 0.1341 | 0.48 | 1.0000 | 0.9201 | 0.7488 | 0.1269 | **_0.1344_** |
| | LIME | 0.34 | | **0.9165** | **0.8000** | **0.1343** | **0.1207** | 0.36 | | **0.9166** | 0.7667 | 0.1327 | 0.1180 |
| 3-vs-8 | SHAP | 0.93 | 0.9859 | 0.4653 | 0.6197 | 0.0786 | **_0.1314_** | 0.29 | 0.9945 | **_0.6197_** | **_0.7261_** | **_0.0996_** | 0.1270 |
| | LIME | 0.41 | | 0.3425 | 0.3803 | 0.0587 | 0.0723 | 0.46 | | **0.4084** | **0.4528** | **0.0689** | **0.0850** |

Table 1: Comparison of two predictive models in terms of proposed metrics, for synthetic dataset and MNIST 3 vs 8 dataset. The best trustworthiness values obtained for each model with LIME are bold-faced, whereas the ones obtained with SHAP are both bold-faced and underlined.

| XAI Method | | AlexNet | | | | ResNet-18 | | | | ResNet-34 | | |
|---|---|---|---|---|---|---|---|---|---|---|---|---|
| | F1 | Fidelity | TWP$_1$ | TWR$_1$ | F1 | Fidelity | TWP$_1$ | TWR$_1$ | F1 | Fidelity | TWP$_1$ | TWR$_1$ |
| SHAP | 0.9670 | 0.88 | **_0.2237_** | **_0.8419_** | 0.9892 | 0.74 | 0.1880 | 0.7678 | 0.9902 | 0.79 | 0.1683 | 0.7258 |
| LIME | | 0.77 | **0.1156** | **0.6847** | | 0.67 | 0.1146 | 0.6464 | | 0.52 | 0.1120 | 0.6425 |

Table 2: Comparison of two predictive models in terms of proposed metrics, for plant disease classification task. Similar bold-facing and underline convention as in Table 1.

(inverse distances) over each ground-truth point. More formally,

$$\text{TWR}_m = \frac{\sum_{g \in \tau_{\mathbf{x}}^m} c' \cdot e^{-d(g,p)}}{\sum_{g \in \tau_{\mathbf{x}}^m} c'}, \text{ where } p \text{ is the closest window in } \phi_k^m \text{ to the window } g. \quad (2)$$

Adopting a similar naming convention, we name this metric as TWR (abbv. for **TrustWorthiness Recall**). Algorithm 1 summarizes the steps of computing TWP and TWR for image domains. For our experiments, we only consider positive SHAP values. This is because negative ones imply that those pixels are contributing negatively towards the prediction but for our ground-truth explanations, we only annotate those regions that should contribute positively towards the prediction.

## 4 EXPERIMENTS

The objective of our experiments is to demonstrate that the proposed metrics provide a complementary dimension of evaluating predictive models in addition to effectiveness, e.g., accuracy, precision, recall etc. A higher value of both TWP, TWR or their harmonic means TWF $=$ $2\text{TWP} \cdot \text{TWR}/(\text{TWP} + \text{TWR})$ (F-score like combination) along with a high accuracy (or F-score) value should be indicative that a model is both trustworthy and effective.

**Synthetic Dataset.** We synthesize a dataset in a way such that the 'gold' explanations can be created with a controlled data generation process. Specifically, we generate a dataset for multi-task image classification wherein the discriminative attributes are - i) the shape type of the figure, and ii) whether the shape is filled or not. In particular, the shape of the objects in our dataset belongs to either of the three following types: a) *triangle*, b) *ellipse* and c) *quadrilateral*. After generating an object of a particular shape, the object is either *filled up* or left unfilled. The dimension of each generated grayscale image is $28 \times 28$. We formulate the problem as a multi-task classification problem, where the task of a predictive model, given an input image, is to classify the shape-type and predict whether the shape is filled or not. The ground-truth explanation for the shape recognition task is the boundary of the shape itself, whereas for predicting if a shape is filled or not, the entire interior region is considered to be the ground-truth. This is illustrated in Figure 3.

**MNIST 3 vs 8 Dataset.** We use a subset of MNIST hand-written digits dataset comprising images from the two classes - 3 and 8, the task being to distinguish between them (Chen et al., 2018). The images of 8 contain two closed loops while those of 3 contain two open loops; hence, the two arcs on the left of the images of 8 that complete the loop are the discriminative regions for 3-vs-8. For ground-truth explanations, we ask the annotators to pick three points on the images of 8, one at the top, one at the middle and one at the bottom, such that when everything to the right of the piece-wise linear curve connecting those 3 points is removed, we end up with those two discriminative arcs. The

| Model (F1) | Metric | GT-1 | GT-2 | GT-3 | GT-4 | GT-5 | Avg |
|---|---|---|---|---|---|---|---|
| BERT+LSTM (0.7036) | TWP | 0.6049 | 0.3964 | 0.4525 | 0.4694 | 0.3646 | **0.4576** |
| | TWR | 0.1709 | 0.1825 | 0.1085 | 0.1504 | 0.1893 | **0.1603** |
| RoBERTa+LSTM (0.7473) | TWP | 0.6015 | 0.3155 | 0.5219 | 0.4532 | 0.3502 | 0.4485 |
| | TWR | 0.1636 | 0.1536 | 0.1351 | 0.1458 | 0.1746 | 0.1545 |

Table 3: Explanation evaluation on the ILDC dataset: the explanations of the hierarchical transformer-based models were obtained with the occlusion-based method Li et al. (2016) on the 5 different versions of the ground-truth annotated by 5 different legal experts. 'Avg' denotes the average values computed over the individual measures for each of the 5 ground-truths.

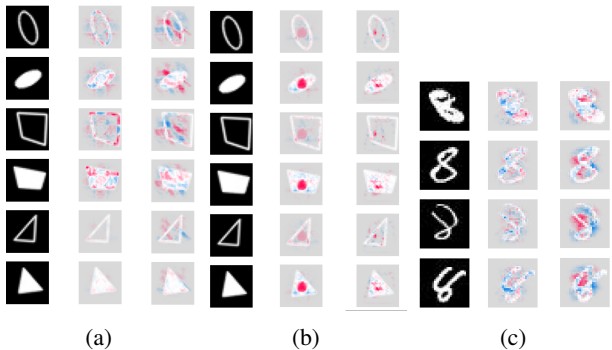

(a)          (b)          (c)

Figure 5: Qualitative model explanations as outputted by SHAP for the multi-task classification on the synthetic dataset and the real datasets. The images, in order from left to right, correspond to - a) 'Shape prediction' SHAP explanation for FF (center) and CONV (right), b) 'Fill prediction' SHAP explanation for FF (center) and CONV (right), and c) SHAP explanation for FF (center) and CONV (right) for the 3-vs-8 classification task.

assignment of images to annotate is done in such a way that one image is annotated by exactly 3 annotators. We use the pixel coordinates of the three points to get the two left arcs as mentioned above. We take all three (from three different annotators) such left arcs and compute the average of them to obtain our final ground-truth explanations, as shown in Figure 2.

It is worth mentioning that defining the ground-truth for instances of some classes is more convenient than defining it for other classes in the same task. For instance, it is easier to define the ground-truth for the class '8' (in terms of the *presence* of a certain region within an image) rather than defining it for the class '3' because in that case, it is not the presence but the *absence* that needs to be considered, which is not so convenient. A total of 70 images of hand-written 8 were annotated.

**Plant-Village Dataset.**   PlantVillage (Mohanty et al., 2016) is a plant disease classification dataset of 54,309 images across 14 plant species and 38 classes (26-12, disease-healthy). The original task is to predict the plant type along with the disease (each combination indicating one class). In such a case, the predictive model should be making use of information both from the patches in the leaves (indicative of disease) along with the discriminative features of the leaves themselves (indicative of the plant type). However, this also means that the explanations should then cater to both the combination of the leaf and the disease types, and as a result they may be more subjective.

To simplify the task, we grouped together the individual classes by the disease types to create 'super-classes', e.g., 'potato early blight' and 'tomato early blight' gets merged into a single class - 'early blight'. Application of this grouping operation along with the removal of classes corresponding to healthy leaves yielded 16 classes in total. This contributed to simplifying the process of manually annotating the ground-truth explanations, because the simplified prediction task should then focus only on the patterns of the leaf patches to identify the correct disease. A sample annotated ground-truth explanation is shown in Figure 4. In total, we annotated 10 images from each class.

**Indian Legal Documents Corpus (ILDC).**   The final dataset that we use for our experiments is the ILDC (Malik et al., 2021) corpus, which is a collection of case proceedings from the Supreme Court of India (SCI). Given a document as an input, the prediction task is to identify the decision of

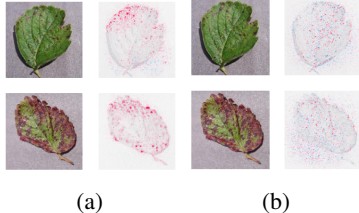

(a)                    (b)

Figure 6: Visualizing sample images from two different ranges of the trustworthiness (TWF) scores on a) AlexNet (F1 = 0.9670) and b) ResNet-34 (F1 = 0.9902). The TWF scores increase as one goes from the top-row to the bottom-row. The row shows a sample image from the set of images where $TWF \in [0.5, 0.75)$, whereas the second row shows a sample where $TWF \geq 0.75$.

| Dataset | FF | | CONV | |
| --- | --- | --- | --- | --- |
| | TWP | TWR | TWP | TWR |
| Syn (Shape) | 0.5070 | 0.7057 | 0.5609 | 0.6589 |
| Syn (Filled) | 0.5814 | 0.6741 | 0.6473 | 0.6611 |
| 3-vs-8 | 0.2990 | 0.6143 | 0.3624 | 0.5624 |

Table 4: Ablation study which treats feature importance uniformly, i.e., when $c$ and $c'$ are set to 1 in Equations 1 and 2, respectively). For these results, the XAI method employed was SHAP.

the judiciary bench (acting as the ground-truth). In addition to the ground-truth decision, a separate set of 56 documents ($ILDC_{expert}$) from the test set documents are annotated by 5 different legal experts. Each annotation constitutes a set of sentences that a human or an AI model should focus on to arrive at the decision.

**Models and Parameter Settings.** The objective in our experiments is to compare the predictive models in terms of their trustworthiness. We do not intend to evaluate the effectiveness of the XAI methods themselves. Therefore, fair comparisons are only to be made between predictive models where the explanations are also obtained with the same XAI methodology, e.g., comparing the TWP or TWR values computed with different explanation methodologies is not fair. In particular, for our experiments we employ two different explanation methodologies - namely SHAP and LIME.

As instances of the black-box model for synthetic dataset and MNIST '3 vs 8' dataset, we employ two different methodologies, namely i) a feed-forward network with one hidden layer (hereafter, referred to as FF model), and ii) 2D convolution layers in addition to the feed-forward ones (hereafter, referred to as CONV model). More specifically, the FF model has one hidden layer of 8 neurons and the CONV model has 2 convolution layers of kernel size 3, the first one having the number of output channels of 3 and the second one 5. The convolution layers are followed by a fully-connected layer with 8 neurons and finally an output layer specific to the task (sigmoid for the 3-vs-8 classifier, whereas a softmax of dimension 3 and a sigmoid for the multi-class objective of shape and fill prediction). For plant disease classification task, we employ AlexNet, ResNet-18 and ResNet-34. We fine-tune the instances of these models pretrained from the ImageNet datset. For ILDC, we use two hierarchical transformer models with RoBERTa and BERT as the transformer units. The [CLS] representations of each text chunk (refer to Malik et al. (2021) for more details on the text chunking) are then fed into BiGRU and then through a dense layer to yield a sigmoid probability. We run our experiments on $ILDC_{multi}$ dataset and use the same hyper-parameter settings as used by the authors. As the explanation method, we use the occlusion method (Li et al., 2016)) as also used in Malik et al. (2021). Our experiments with LIME used the number of image segments as 20 for the synthetic and MNIST 3 vs 8 datasets, whereas this was set to 1000 for the plant disease classification task owing to the larger image sizes.

**Results.** To find how faithfully the local explanation models approximate the primary models, we randomly sample 100 instances from the test set and compute the difference of accuracy between the original predictions and the new predictions after masking out the top 25% important features (as ranked by by explanation method). These fidelity scores (Pope et al., 2019) are reported in Tables 1 and 2. They show that the difference in accuracy is always $\geq 29\%$, thereby establishing

the faithfulness of the XAI models - LIME and SHAP for the tasks in our experiments. Figure 5 shows the pixel-level explanations as outputted by SHAP for both FF and CONV models, for the synthetic dataset. It can be noted that both the models attend to the interiors of the shapes for task 2 (fill prediction). But when it comes to task 1 (shape prediction), the FF model seems to be attending to the relevant pixels better than the CONV model. Specifically, for the fill prediction task, a majority of the high intensity red points (high SHAP values) lie on the circumference or in close proximity of it. However, for the shape prediction task, such points are spread out farther away from the human-interpretable regions. Our proposed metrics are able to capture this behaviour as depicted in Table 1. It should be noted that despite the TWP and TWR values being slightly different when we use SHAP versus LIME, the relative rank of the models based on the performance with respect to these metrics is still the same (compare TWP and TWR columns between the consecutive rows, e.g., rows 1 and 2 etc.).

An important observation worth noting is that despite having higher accuracy, the CONV model fails to attend to the human-interpretable regions within the images. Therefore, somewhat surprisingly, the FF model's way of reasoning is more in parity with that of a human in comparison to a 2D convolutional network.

Figure 5 shows the pixel-level explanations as outputted by SHAP for both FF and CONV models, for a real dataset. It can be noted that the CONV model assigns a substantially higher importance to the two left arcs (see Figure 2) than the FF model. Therefore, in this task, the abstractions from the data that CONV learns agrees more with that of human's than is the case for FF. Unlike the synthetic dataset, here the model with higher accuracy attends to the human-perceived discriminatory regions in a better manner as compared to the lower accuracy one. This is again reflected in our proposed metrics, as depicted in Table 1, regardless of the choice of the XAI model.

Figure 6 shows the pixel-level explanations as outputted by SHAP for AlexNet and ResNet-34 models, on the plant disease classification task. It can be observed that for predictions yielding the highest TWF values (the bottom-most row), AlexNet shows a better agreement with the ground-truth than ResNet-34, as can be seen from the presence of a considerable quantity of red regions (i.e., where the model focuses most) **even outside the leaf**. This shows that a model with a higher effectiveness may not necessarily be attending to a set of data units that a human would focus on. This is also seen for the top row of Figure 6, where we see that a model with lower effectiveness (AlexNet) despite showing the presence of attention regions outside the leaf (thus the lower TWF scores) shows a denser distribution of focus regions inside the leaf nonetheless. In contrast, in ResNet-34 the distribution of focus regions is not only sparse but even outside the leaf.

Table 3 shows the comparison of TWP and TWR scores for the hierarchical transformer model explanations obtained using occlusion method with 5 different ground-truth annotations, for ILDC dataset. Observations similar to the plant disease classification task can also be made for the Court Judgement Prediction and Explanation (CJPE) task, i.e., a model with lower effectiveness (BERT+LSTM) yields higher trustworthiness scores (TWP and TWR).

**Ablation Study.** We now measure the metrics TWP and TWR without using identical feature importance, and see if this is able to produce a reversal in the relative trustworthiness estimation across models. In other words, the $\phi_k$ set in this setup now comprises elements of the form $(h, w, 1)$, i.e., the importance $c$ of each pixel is set to a uniform value of $1$. The results of this ablation, using SHAP as the fixed XAI method, is shown in Table 4. We observe that, contrary to the true metric values of Table 1, the TWR values in Table 4 of the CONV model on the 3-vs-8 classification task is worse than FF, which shows a reversal of the relative ordering of trustworthiness. This, in turn, shows that taking into account the feature weights themselves as obtained from an explainer model is indeed important.

**Concluding Remarks.** In this study, we proposed a general approach to quantifying the trustworthiness of a predictive model and apply it on a synthetic dataset and three real datasets. We proposed two new metrics for the quantitative evaluation of model trustworthiness, and demonstrated that the model with a higher effectiveness may not necessarily be attending to the most human-interpretable features, thereby hinting towards the importance of looking beyond model effectiveness measures, such as accuracy, precision, recall etc.

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
