# OpenReview forum: "TOWARDS AN OBJECTIVE EVALUATION OF THE TRUSTWORTHINESS OF CLASSIFIERS"
_ICLR.cc/2023/Conference — Submitted to ICLR 2023_

### Official Review · Reviewer_KrGW · 2022-10-17

**Confidence:** 4
**Correctness:** 1
**Technical Novelty And Significance:** 2
**Empirical Novelty And Significance:** 1
**Recommendation:** 1

**Clarity, Quality, Novelty And Reproducibility:**

While the topic raised in this work is important and interesting, the proposed model suffers from critical drawbacks and logical flaws.

The main goal of this work is to develop an algorithm and a set of metrics to evaluate the trustworthiness of a model (independent of XAI's trustworthiness). This work uses XAI tools (SHAP and LIME) as an intermediate step for trustworthiness evaluation. It is not obvious that a model with low TWP and TWR is indeed untrustworthy or that the XAI methods produced unacceptable explanations, i.e. the  XAI is untrustworthy.

To address this drawback, this work argues that: "The objective in our experiments is to compare the predictive models in terms of their trustworthiness. We do not intend to evaluate the effectiveness of the XAI methods themselves. Therefore, fair comparisons are only to be made between predictive models where the explanations are also obtained with the same XAI methodology, e.g., comparing the TWP or TWR values computed with different explanation methodologies is not fair."
Now, suppose using SHAP, one finds model A has AUC = 0.9 while TWP and TWR are below 0.6, and model B has AUC = 0.6 while TWP and TWR are above 0.9. Should we trust the second model more than the first model? Suppose there is a model with AUC = 0.99 (MNIST) but TWP and TWR are below 0.6 while for another model with AUC = 0.95, TWP and TWR are 0.8, is this guaranteed that the second model is more trustworthy than the first model? An XAI method can easily fail in producing a good explanation and then using the proposed method one might conclude that a model is untrustworthy which is incorrect. This is actually not a hypothetical situation, the authors already have shown this in Table 2.

The authors correctly mentioned that the trustworthiness of a model should be independent of the XAI method, but the current method cannot distinguish between an untrustworthy model and when the XAI methods are untrustworthy (not the actual model).

It is also sensitive to annotation. Another important problem is with human annotation. The regions annotated by humans as the regions of interest might not actually be the most optimal regions to differentiate between two classes (for example in distinguishing between 3 and 8). A model that looks for two connected closed loops to distinguish between 3 and 8, is a good model, a trustworthy model, and can achieve the goal of distinguishing between 3 and 8. There is no need for a trustworthy model to find the pixel level difference between 3 and 8. Hence, the human annotation that is used as a reference point is subjective, while the trustworthiness of a model should not be subjective.

**Strength And Weaknesses:**

Strengths:

-- Interesting and important topic. As rightly pointed out having XAI methods are not sufficient to argue whether a model is trustworthy or not.
-- This work is not about evaluating XAI models, it is about the evaluation of a trained ML model. This adds a new dimension to XAI literature.

Weaknesses:

-- there are critical logical flaws in the arguments and models.
-- the models developed in this work do not solve the problem raised in the motivation.

**Summary Of The Paper:**

This work develops a framework to evaluate the trustworthiness of a model using XAI tools and TWP, TWR that are developed in this work. Then, it illustrates an application of their framework to several real-world settings.

**Summary Of The Review:**

This work has important logical flaws and I cannot recommend this work for publication even after a major review.

---

> ### Author Response · Authors · 2022-11-18
> **We have computed the fidelity scores to establish the trustworthiness of the XAI methods.**
>
> 1. ...this guaranteed that the second model is more trust-worthy than the first model? An XAI method can easily fail in producing a good explanation and then using the proposed method one might conclude that a model is untrustworthy which is incorrect...
>     We have computed fidelity scores as a measure of the local fits of the explanation models (reported in Tables 1 and 2). The results show considerable high fidelity scores indicating a good local approximation of the complex black-box models. Looking at both the fidelity and the TWP/TWR measures could be an indication of the overall fidelity (closeness of approximation) and trustworthiness (correlation with human judgments for the decisions).
> 2. The authors correctly mentioned that the trustworthiness of a model should be independent of the XAI method, but the current method...
>     This recursive argument definitely opens up scopes for philosophical discussions; however, from a practical point-of-view, we have reported the fidelity scores in Tables 1 and 2, which are sufficiently high to show that the standard explanation models, e.g., LIME and SHAP do a fair job in locally approximating a complex model in our experiment settings.
> 3. It is also sensitive to annotation.
>     Any data-driven model is also sensitive to annotations; so, we don't think this qualifies to be a flaw, as such.
> 4. Another important problem is.........Hence, the human annotation that is used as a reference point is subjective, while the trustworthiness of a model should not be subjective.
>     Our metric precisely quantifies what the reviewer has said, i.e., to "check for the connected closed loops to distinguish between 3 and 8". Instead of the closed loops, we only consider the semi loops on the left because that's the discriminating feature between 3 and 8.. If the explanation weights for a prediction model isn't showing high attention over these semi loops then the model is less likely to be trustable, and that's precisely what we report in our measures.

---

### Official Review · Reviewer_G8bv · 2022-10-23

**Confidence:** 4
**Correctness:** 2
**Technical Novelty And Significance:** 1
**Empirical Novelty And Significance:** 1
**Recommendation:** 3

**Clarity, Quality, Novelty And Reproducibility:**

Paper is clearly written but neither the quality nor the originality suffice for publication in ICLR. Since the authors have not provided the codes, the results are not reproducible.

**Strength And Weaknesses:**

Strength

It is quite important to come up with a measure for trustworthiness of trained models. The authors have motivated their study nicely, and presented their measures in a clear manner.

Weaknesses

Both measures are straightforward. This could be an advantage, but this also causes them to overly simplify the concept of providing explanations, or for that matter, measuring trust. The ground-truth explanations for images are obtained from experts. There could be a few concerns there: First, creating a binary matrix for an image is a tedious task that is prone to errors. Second, when looking at an image, the humans look at the pattern that are caused by a collection of pixels not just single points (e.g., a skin lesion where the doctors look at the symmetry or the size of the lesion). When presented with same images that are transformed (e.g., rotated) the ground-truth would completely change.

A few questions for the authors:

1. SHAP also returns negative weights. In that case, do you use the absolute values of these weights as the feature importance?

2. As authors clearly stated on page 6, "[...] it is not the presence but the absence that needs to be considered [...]" This is very important, and I believe cannot be addressed with the current measures. Am I right?

3. Both measures are obtained by averaging over all samples in the evaluation set. Could the sample variance or the outliers would be of more interest than averages?

**Summary Of The Paper:**

This paper proposes a quantitative measure of trustworthiness of a black-box model trained on text or image datasets. This measure requires a set of ground-truth explanations to compare them against explanations provided by an explainable artificial intelligence (XAI) method applied to a black-model model. After introducing their measure, the authors test their measure on a set of datasets.

**Summary Of The Review:**

The authors have tried to provide a solution to a very difficult problem. Both proposed measures, in their current states, are only ad-hoc solutions.

---

> ### Author Response · Authors · 2022-11-18
> **We have now generalized the algorithm by looking at m*m windows at a time, rather than a single pixel.**
>
> 1. Both measures are straightforward. This could be an advantage, but this ...
>     We have generalized the metrics now to operate at different levels of granularity. The details are in algorithm 1.
> 2. The ground-truth explanations for images are obtained from experts. There could be a few concerns ...
>     We actually end up having a binary matrix, but during annotation, we do it at superpixels (regions) level, thereby reducing the chances of errors. Secondly, we have now made the algorithm more general by looking at m*m windows at a time, rather than a single pixel. We have reported the numbers in Tables 1 and 2.
> 3. SHAP also returns negative weights. In that case, do you use the absolute values ...
>     For our experiments, we only consider positive SHAP values. This is because negative ones imply that those pixels are contributing negatively towards the prediction but for our ground-truth explanations, we only annotate those regions that should contribute positively towards the prediction. We have not included this piece of information in section 3 paragraph ``Recall measure''.
> 4. As authors clearly stated on page 6, "[...] it is not the presence but the absence that ...
>     The proposed metrics can still be computed, but it's the labelling of GTE that's the bottleneck.
> 5. Both measures are obtained by averaging over all samples ...
>     One can also consider getting Geometric Mean instead of Arithmetic Mean.
> 6. Since the authors have not provided the codes, the results are not reproducible.
>     Due to a system error, the zip files were corrupted. We later uploaded the same zip file again after being notified from the track chairs.

---

### Official Review · Reviewer_isth · 2022-10-24

**Confidence:** 2
**Clarity, Quality, Novelty And Reproducibility:** All of the above are satisfactory
**Correctness:** 4
**Technical Novelty And Significance:** 3
**Empirical Novelty And Significance:** 3
**Recommendation:** 8

**Strength And Weaknesses:**

+ The paper is well motivated and well written
+ Recent work seems to have been discussed
+ The measures are simple but intuitive
+ Experiments are conducted on real datasets

- It would be nice if the measures can be incorporated to produce both trustworthy and effective clusters

**Summary Of The Paper:**

The authors provide a quantitative measure to check the correspondence of feature importance as detected by a model with human perception of a feature's importance, which is incorporated in a framework to quantify model trustworthiness.

**Summary Of The Review:**

I enjoyed reading the paper because it addressed the problem with clarity, and due to the simplicity of the defined measures

---

> ### Author Response · Authors · 2022-11-18
> **Code zip file was corrupted due to a system error, we were asked to resubmit it later.**
>
> Due to a system error, the zip files were corrupted. We later uploaded the same zip file again after being notified from the track chairs.

---

### Official Review · Reviewer_8Lv7 · 2022-10-24

**Confidence:** 4
**Correctness:** 2
**Technical Novelty And Significance:** 2
**Empirical Novelty And Significance:** 2
**Recommendation:** 5

**Clarity, Quality, Novelty And Reproducibility:**

The paper is written clearly. In terms of writing and some performed experiments, the paper can be considered as somewhat high quality; however, in terms of technical rigor and proposed framework and metric more discussions need to take place. The authors mentioned that data and code will be available for reproducibility but at the moment no such material is realsed.

**Strength And Weaknesses:**

**Strengths:**
1. The problem considered in this paper is interesting.
2. The framework is applicable to text and image domain.
3. Paper is well-written and easy to follow.
4. The collected datasets in this paper can be useful to the community.

**Weaknesses:**
1. The approach needs ground truth explanation collection for each task which I am not sure how feasible might be.
2. In some cases this ground truth collection might not be easy in cases where expert knowledge is required. What do authors think about these types of overheads. Please discuss.
3. The paper is not technically rigor and am not sure about the extend of novelty introduced in this work.
4. Authors (and ultimately the metric and framework proposed in this work) assume that the explanation methods are themselves complete and reliable such that the measures are being build upon these explanation methods. More discussion and justification needs on this issue.

**Summary Of The Paper:**

The paper proposes a new framework for measuring trustworthiness of a model including new proposed metrics that can evaluate various models comparingly. The framework is applicable to text and image domains and is validated through experiments on datasets from each domain accordingly. The results showed that a model with higher accuracy might not be trustworthy according to the proposed metrics and framework introduced in the paper.

**Summary Of The Review:**

Overall, the problem considered is interesting, but the paper has some flaws in terms of lacking some more in depth discussions along with technicality.

---

> ### Author Response · Authors · 2022-11-18
> **We have computed fidelity scores to establish the faithfulness of the XAI methods.**
>
> 1. The approach needs ground truth explanation collection ...
>     Without manually created human annotations, user studies are anyway needed, which requires even more manual effort because new user studies need to be done for newer explanation and black box models.
> 2. In some cases this ground truth collection might not be easy ...
>     GTE collection for some tasks can be difficult, as is the case for standard ML tasks where the labeling process is subjective; for the time being, we focus on tasks that are not too subjective in nature.
> 3. The paper is not technically rigor and ...
>     This is the first attempt to quantify the explainability of black-box models.
> 4. Authors (and ultimately the metric and framework proposed in this work) assume that ...
>     We have computed fidelity scores as a measure of the local fits of the explanation models (reported in Tables 1 and 2). The results show considerable high fidelity scores indicating a good local approximation of the complex black-box models.
> 5. The authors mentioned that data and code will be available for reproducibility but at the moment...
>     Due to a system error, the zip files were corrupted. We later uploaded the same zip file again after being notified from the track chairs.

---

> > ### Comment · Reviewer_8Lv7 · 2022-12-03
> > **Response to Authors**
> >
> > Thank you for providing the responses. I appreciate the time and effort you put in providing responses to reviewers. I am still not 100% sold on the response given to my ground truth requirement concern. In addition, the technical rigor is another major concern. The idea is interesting; however, more effort needs to be taken on providing more in depth justification on the concerns raised in my review.

---

### Decision · Program_Chairs · 2023-01-20

**Decision:**

Reject

**Justification For Why Not Higher Score:**

There are major concerns in the rigorousness of the work, overly simplifying the concept of measuring trust, the sensitivity/feasibility of the framework for getting annotations, etc.


**Justification For Why Not Lower Score:**

N/A

**Metareview: Summary, Strengths And Weaknesses:**

Summary:
The authors proposed a general framework to quantify the trustworthiness of a black-box model on image or text datasets.

Strength:
Paper is well-written. This is an important topic to measure the trustworthiness of the models.

Weakness:
There are major concerns in the rigorousness of the work, overly simplifying the concept of measuring trust, the sensitivity/feasibility of the framework for getting annotations, etc.


**Summary Of Ac-Reviewer Meeting:**

N/A